# Beneficial effects of resistance training on both mild and severe mouse dystrophic muscle function as a preclinical option for Duchenne muscular dystrophy

Medhi Hassani[1], Dylan Moutachi[2], Mégane Lemaitre[3], Alexis Boulinguiez[4], Denis Furling[2], Onnik Agbulut[1], Arnaud Ferry[2,5]*

1 Sorbonne Université, Institut de Biologie Paris-Seine, UMR CNRS 8256, INSERM ERL U1164, Biological Adaptation and Ageing, Paris, F-75013 France, 2 Sorbonne Université, INSERM, Institut de Myologie, Centre de Recherche en Myologie, F-75013 Paris, France, 3 Sorbonne Université, UMS28, Paris, F-75013 France, 4 Department of Biological Sciences, Royal Holloway University of London, Surrey, United Kingdom, 5 Université Paris Cité, F-75006 Paris, France

* arnaud.ferry@upmc.fr

**Data Availability Statement:** All relevant data are within the paper and its Supporting Information files.

## Abstract

Mechanical overloading (OVL) resulting from the ablation of muscle agonists, a supra-physiological model of resistance training, reduces skeletal muscle fragility, i.e. the immediate maximal force drop following lengthening contractions, and increases maximal force production, in *mdx* mice, a murine model of Duchene muscular dystrophy (DMD). Here, we further analyzed these beneficial effects of OVL by determining whether they were blocked by cyclosporin, an inhibitor of the calcineurin pathway, and whether there were also observed in the D2-mdx mice, a more severe murine DMD model. We found that cyclosporin did not block the beneficial effect of 1-month OVL on plantaris muscle fragility in *mdx* mice, nor did it limit the increases in maximal force and muscle weight (an index of hypertrophy). Fragility and maximal force were also ameliorated by OVL in the plantaris muscle of D2-*mdx* mice. In addition, OVL increased the expression of utrophin, cytoplamic γ-actin, MyoD, and p-Akt in the D2-*mdx* mice, proteins playing an important role in fragility, maximal force gain and muscle growth. In conclusion, OVL reduced fragility and increased maximal force in the more frequently used mild *mdx* model but also in D2-*mdx* mice, a severe model of DMD, closer to human physiopathology. Moreover, these beneficial effects of OVL did not seem to be related to the activation of the calcineurin pathway. Thus, this preclinical study suggests that resistance training could have a potential benefit in the improvement of the quality of life of DMD patients.

## Introduction

Lengthening contractions, for example when the muscle slows down the movement, are known to cause immediate and prolonged reduction in skeletal muscle function and delayed

**Funding:** The author(s) received no specific funding for this work.

**Competing interests:** The authors have declared that no competing interests exist.

muscle soreness, even in healthy muscle. The skeletal muscle in murine Duchenne muscular dystrophy (DMD) models, with dystrophin deficiency, is more susceptible to lengthening contraction-induced loss of function compared to healthy muscle, i.e., more fragile. More precisely, it is the fast and low oxidative muscle that is more susceptible to lengthening contraction-induced loss of function in the mild *mdx* murine DMD model, in contrast to the slow and more oxidative muscle which is not more fragile [1, 2]. Certain proteins, such as utrophin, cytoplamic γ-actin as well as desmin, involved in muscle structure, are also known to significantly modulate this increased fragility when dystrophin is absent. Indeed, the overexpression of utrophin [3] or cytoplamic γ-actin [4] markedly reduces the fragility of *mdx* mice, while the absence of desmin increases it [5]. Genetic activation of the calcineurin and Akt pathways can also protect the murine dystrophic muscle from susceptibility to lengthening contraction-induced loss of function whereas genetic inactivation of the mTOR pathway increases it in healthy muscle [6–9]. It was also recently reported that overexpression of the myogenic transcription factor MyoD also aggravates muscle fragility [10]. These pathways are also known to regulate fiber-type specification and growth, and play a role in the adaptations induced by chronic muscular exercise in healthy muscle [11–15].

Interestingly, chronic muscular exercise can decrease the susceptibility to lengthening contraction-induced loss of function in *mdx* mice. Chronic voluntary running and low-frequency electrical stimulation, likely animal models of endurance training, reduce fragility in *mdx* mouse fast muscle [16–18], whereas physical inactivity aggravates it [16]. Recently, it was found that the voluntary running-induced improvement of *mdx* mice muscle fragility was related to the activation of the calcineurin pathway, and associated with changes in the program of genes involved in slower contractile phenotype of the muscle fibre [17]. In this latest study, these beneficial effects of calcineurin were blocked by administration of cyclosporin (CsA), a calcineurin inhibitor [17]. In addition, it has been shown that murine models of resistance training also reduce fragility in the *mdx* mouse [19, 20]. In fact, 6 resistance training sessions [19] and chronic mechanical overloading (OVL) of the plantaris muscle [20], a supraphysiological model of resistance training [21], reduce the force drop following lengthening contractions in the lower leg muscles of *mdx* mice [19, 20]. However, it is not yet known whether CsA blocks the beneficial effect of resistance training on fragility in dystrophic muscle. Moreover, since murine models of resistance training also increase muscle weight (an index of hypertrophy) and maximal force in *mdx* mice [19, 20, 22], it would be interesting to know whether these maximal force and weight gains are blocked by CsA, suggesting or not a role of the calcineurin pathway in these beneficial effects. CsA has been reported to decrease the hypertrophy in OVL healthy muscle [23, 24], although this is not found by all [11]. To our knowledge, the contribution of the calcineurin pathway to the strength gain induced by resistance training is not yet known either in healthy or dystrophic muscles.

The previous studies investigating chronic resistance-like muscular exercise did not use a severe murine DMD model, with a major functional deficit closer to that of the DMD patient. A recently developed murine model for DMD is the D2-*mdx* mice that exhibits a more marked weakness (reduced absolute maximal force) compared with *mdx* mice since there is no muscle hypertrophy in D2-*mdx* mice contrary to *mdx* mice [25–27]. Thus, it would be relevant to determine whether a model of resistance training also improves the fragility in the D2-*mdx* mice, and likewise the absolute maximal force.

The purpose of the present study was to further analyze the beneficial effect of OVL on muscle fragility and absolute maximal force in dystrophin deficient mice, two important functional dystrophic features. In particular, we wanted to know if the administration of CsA, known to inhibit the calcineurin signaling pathway, blocked the beneficial effect of OVL on fragility of *mdx* dystrophic muscle, as it did in chronic running exercise [17]. We also wanted

to determine whether CsA blocked the effect of OVL on maximal force gain in dystrophic muscle since CsA has been reported to inhibit hypertrophy at least in healthy muscle and possibly the maximal force gain [23, 24]. Another important specific aim was to determine if OVL also exerted these beneficial effects in a more severe murine model of DMD, the D2- *mdx* mice.

## Materials and methods

### Ethical approval and animal models

Animal were housed in the departmental animal facility, with free access to water and rodent laboratory chow. The animal facility is specific pathogen free, with a 12h-light/12h-dark cycle, and mice were 3–5 per cage. All procedures were performed in accordance with national and European legislations and were approved by the French Ministere de l'Enseignement Supérieur de la Recherche et de l'Innovation (APAFIS #21554–2019071912051421). Male *mdx* mice (with hybrid background C57Bl/6 x C57Bl/10) and D2- *mdx* (DBA2/J background) mice were bred locally and were used at 3–6 months of age.

### Experimental design

Mice were randomly divided into different age-matched control and experimental groups. We performed 3 separate experiments (Fig 1) to determine: 1) whether the administration of CsA blocked the effects of OVL on plantaris muscle in *mdx* mice, with the ablation of soleus and the major portion of lateral and medial gastrocnemius muscles, 2) the effects of OVL on lateral gastrocnemius muscle in *mdx* mice, with the ablation of a smaller weight of agonist muscles, a less severe OVL, and 3) the effects of OVL on plantaris muscle in D2-*mdx* mice, a much severe murine DMD model compared to *mdx* mice. In the second experiment, we wanted to induce a lower mechanical overload compared to that imposed on the plantaris muscle since the OVL of the plantaris muscle is supra-physiological, and leads to strength and mass gains approaching 100%, rarely seen in humans [21, 28]. Muscles were studied 1 month after the initiation of OVL.

### Mechanical overloading

For mechanical overloading (OVL) in the 3 experiments, dystrophic mice were anaesthetized with isoflurane. The plantaris muscles (Experiments 1 and 3) of both legs were mechanically overloaded (Mdx+OVL, Mdx+OVL+CsA, and D2-mdx+OVL mice) for 1 month by surgical removal of soleus muscles as well as a major portion of the lateral and medial gastrocnemius muscles as described [20, 22] (Fig 1). The lateral gastrocnemius muscles (Experiment 2) of both legs were mechanically overloaded (Mdx+OVL) for 1 month by surgical removal of soleus, plantaris muscles as well as a major portion of the medial gastrocnemius muscles (Fig 1). Muscles were measured and collected 1 month after the initiation of OVL.

### Cyclosporin A (CsA) treatment

In experiment 1, mice were treated every day during 1 month with the calcineurin pathway inhibitor cyclosporine A, CsA (25 mg/kg, ip, daily)(Mdx+OVL+CsA mice)[17, 23, 24] or saline (Mdx+OVL and Mdx mice) (Fig 1), the day after muscle ablation.

### Muscle weakness and fragility

In the 3 experiments, maximal force and fragility (susceptibility to contraction-induced functional loss) were evaluated by measuring the *in situ* muscle contraction in response to nerve

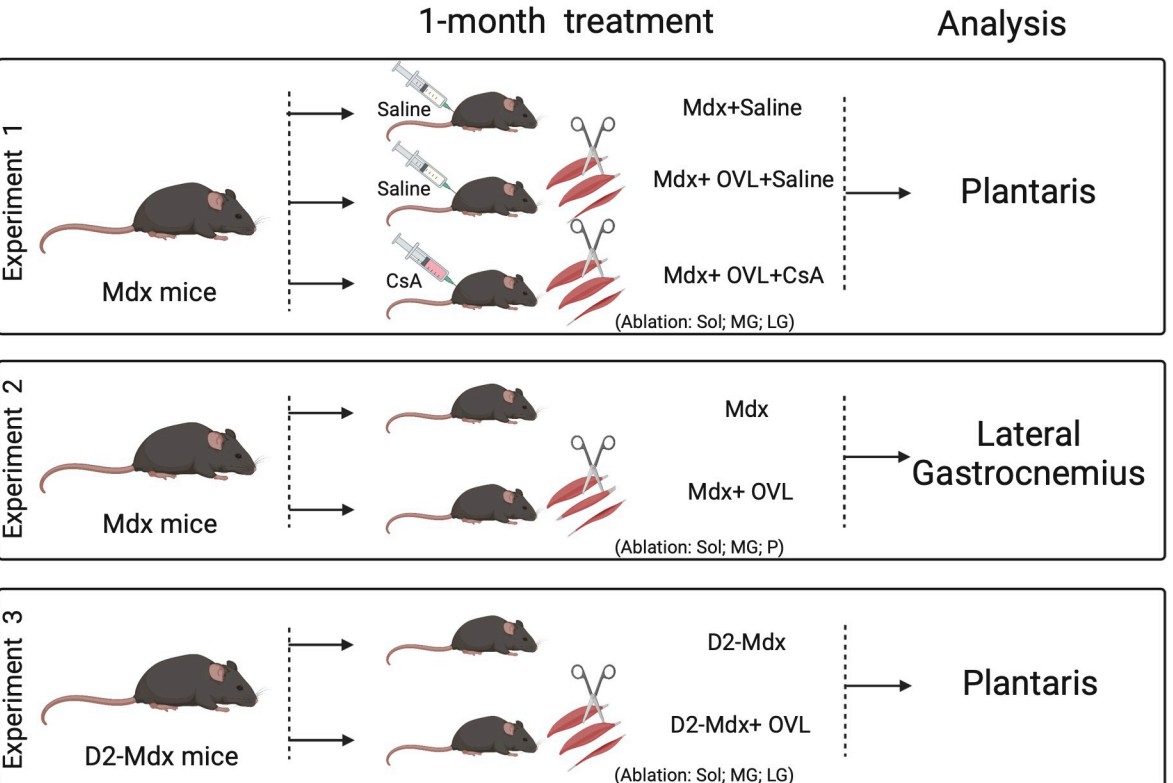

**Fig 1.** Experimental design of the study, 3 experiments were performed to determine: 1) whether the administration of CsA blocked the effects of OVL on plantaris muscle in Mdx mice (experiment 1), 2) the effects of OVL on lateral gastrocnemius muscle in Mdx mice (experiment 2), a lower OVL, and 3) the effects of OVL on plantaris muscle in D2-mdx mice, a more severe murine DMD model compared to mdx mice. OVL: mechanical overloading.CsA: cyclosporin A. Mdx+OVL: mechanically overloaded Mdx mice. Mdx+OVL+CsA: mechanically overloaded Mdx mice that received *CsA*. Mdx: non-overloaded Mdx muscle. D2-mdx+OVL: mechanically overloaded D2-mdx mice. D2-mdx: non-overloaded D2-mdx muscle.

stimulation, as described previously [20]. Mice were anesthetized using pentobarbital (60 mg/kg, ip). Body temperature was maintained at 37°C using radiant heat. The knee and foot were fixed with pins and clamps and the distal tendon of the muscle was attached to a lever arm of a servomotor system (305B, Dual-Mode Lever, Aurora Scientific) using a silk ligature. The sciatic nerve was proximally crushed and distally stimulated by a bipolar silver electrode using supramaximal square wave pulses of 0.1 ms duration. We measured the absolute maximal force that was generated during isometric contractions in response to electrical stimulation (frequency of 75–150 Hz, train of stimulation of 500 ms). Absolute maximal force was determined at L0 (length at which maximal tension was obtained during the tetanus). L0 was measured with a caliper (distal tendon length was not included). The ability to rapidly generate muscle force was also assessed in Experiment 1. The rate of force development (RFD)(g/s) was measured when the force increased from 0 to 25% (RFD0-25%), 25% to 50% (RFD25-50%) and 50% to 75% (RFD50-75%) of P0 because the mechanisms responsible for RFD vary over time [29].

Fragility, i.e., susceptibility to contraction-induced functional loss in mice was estimated from the immediate force drop resulting from lengthening contractions. The sciatic nerve was stimulated for 700 ms (frequency of 150 Hz). A maximal isometric contraction of the muscle was initiated during the first 500 ms. Then, muscle lengthening (10% L0) at a velocity of 5.5 mm/s was imposed during the last 200 ms. All isometric contractions were made at an initial

length L0. Nine lengthening contractions of the plantaris muscles were performed in mice, each separated by a 60-s rest period. Maximal isometric force was measured 1 min after each lengthening contraction and expressed as a percentage of the initial maximal force. After contractile measurements, the animals were killed with cervical dislocation.

## Protein levels

Muscles from the Experiment 3 were snap-frozen in liquid nitrogen immediately after dissection. Frozen muscles were homogenized by Ultra Turax into an ice-cold in Newcastle Buffer [30] containing: 75 mM Tris pH 6.8, 3.8% SDS, 4 M urea, 2 M thiourea, 20% glycerol and 1% of protease inhibitor cocktail (Thermo Scientific) for immunoblotting. Samples were, incubated 30 min in ice and then centrifuged at 4°C. Protein concentration was measured using the DC Protein ASSAY (BioRAD) method with bovine serum albumin as a standard. Equal amounts of protein extracts (30 μg) were distributed and separated in Gel 5/15% or Gel 4/7.5% (Mini-Protean TGX Precast Gels, BioRAD) before electrophoretic transfer onto a 0.2 μm nitrocellulose blotting membrane (Amersham Protran). Western-blot analysis was conducted using anti-Utrophin (anti-Mouse; DSHB), anti-Desmin [Y66] (anti-Rabbit; Abcam), anti-fast myosin heavy chain isoform, MHC2a [SC-71] (anti-Mouse; DSHB), anti-AKT (anti-Rabbit; Cell Signaling), anti-P-AKT (Ser473) (anti-Rabbit; Cell Signaling), anti-cytoplamic γ-actin (anti-Mouse; SIGMA), anti-MyoD (anti-Mouse; Santa Cruz), anti-oxidative phosphorylation complex cocktail (OXPHOS) (anti-Mouse; Abcam), and anti-HSP-60 (anti-Goat, Santa Cruz) antibodies. Proteins bound to primary antibodies were visualized with secondary antibodies IRDye® 680LT Donkey anti-Goat or IRDye® 800LT Donkey anti-Goat (LI-COR), or HRP-conjugated anti-Mouse or HRP-conjugated anti-Rabbit secondary antibodies (Biorad). The signals were detected using ECL reagent (Cyanagen) then revealed with Chemidoc. Bands were quantified by Image Studio Lite software. The ratios of utrophin, desmin, cytoplamic γ-actin, MHC-2a, oxidative phosphorylation complexes, Akt, p-Akt, MyoD, to HSP-60 were calculated and then normalized to Mdx values.

## Statistical analysis

Statistical analyses were performed Prism v8.4.0 software (GraphPad, La Jolla, CA, USA). When comparing two groups of data for one variable, T-test, T-test with Welch's correction (when variance between groups was different) or Mann-Whitney test (when the number of data per group ≤ 5, typically protein levels) was used. When comparing three groups of data for one variable, one-way ANOVA with Tukey test or Brown-Forsythe ANOVA with Dunnet T3 test (when variance between groups was different) was used. Variance difference between group was tested using Brown-Forsythe test. When comparing data groups for more than one variable (typically the force drop following lengthening contractions in the 3 experiments), two-way ANOVA and Sidak test (when there was an interaction between factors) was used. We also calculated the Pearson correlation coefficients. P value less than 0.05 was considered significant. We did not remove any outlier. Individual values are presented in graphs, mean and SEM.

## Results

### CsA did not block the OVL-reduced fragility and maximal force and weight gains in Mdx mice (Experiment 1)

*In situ* isometric and lengthening contractions in response to nerve stimulation in the OVL plantaris muscle were performed 1 month after muscle agonist ablation (Experiment 1, Fig 1).

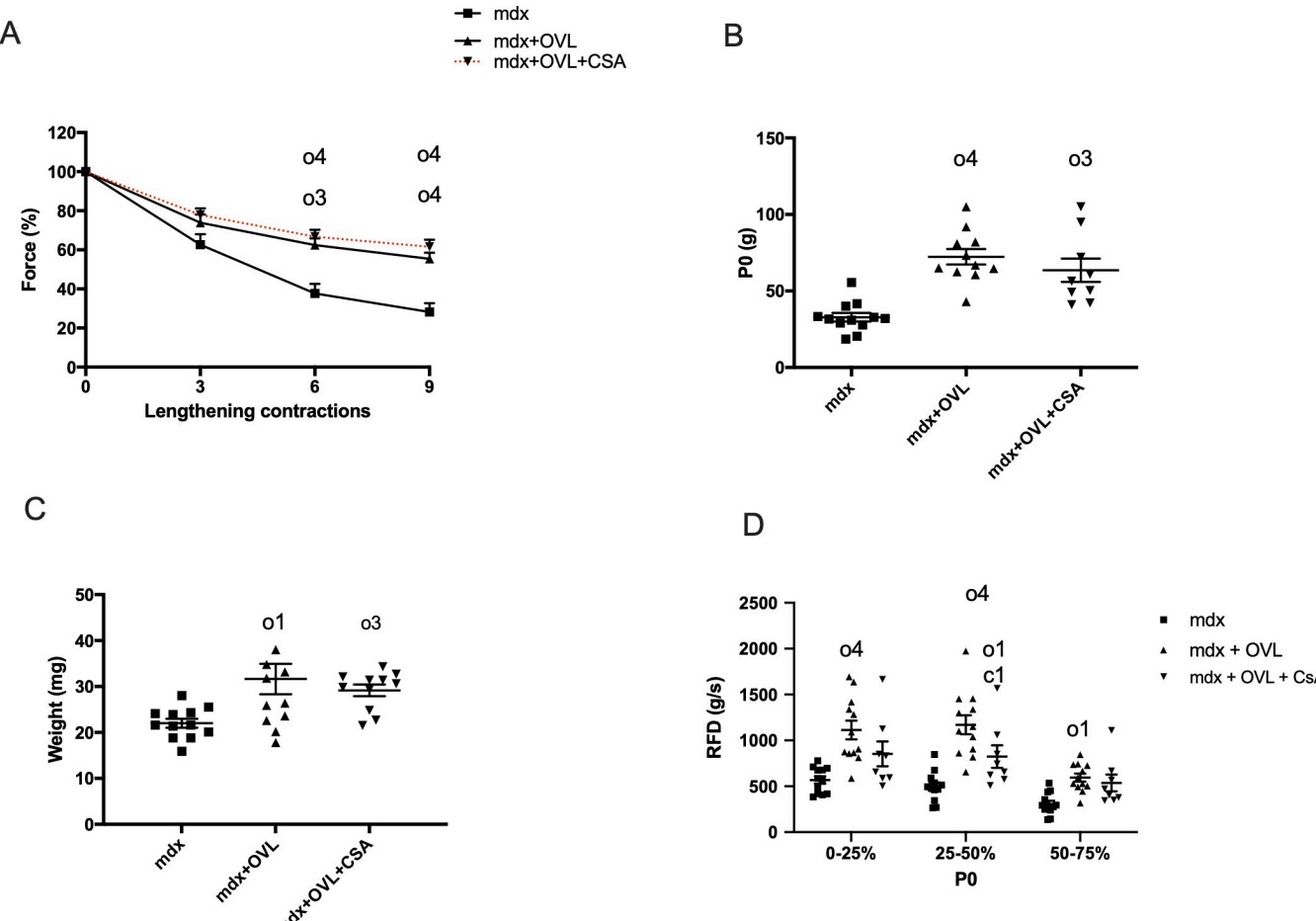

**Fig 2. Effect of CsA on OVL plantaris muscle in Mdx mice.** (A) Force drop following lengthening contractions (Fragility). n = 8–12 per group. (B) Absolute maximal force (P0). n = 9–12 per group. (C) Muscle weight. n = 6–8 per group n = 11–12 per group. (D) Rate of force development. n = 9–12 per group. CsA: cyclosporin A. Mdx+OVL: mechanically overloaded Mdx mice. Mdx+OVL+CsA: mechanically overloaded Mdx mice that received *CsA*. Mdx: non-overloaded Mdx muscle. o1, o3, and o4: significant different from Mdx (p < 0.05), (p < 0.001) and (p < 0.0001) respectively. c1: significant different from Mdx+OVL (p < 0.05).

As previously observed [20], the isometric maximal force drops following lengthening contractions were reduced in Mdx+OVL mice compared with Mdx mice (Fig 2A), indicating a reduced fragility in Mdx+OVL. In fact, the remaining force following the 9th lengthening contraction was increased in Mdx+OVL mice compared with Mdx mice (55.4% versus 28.2%)(p< 0.0001) (Fig 2A). We also found that absolute maximal force (+120.4%)(p < 0.0001) (Fig 2B) and muscle weight (+ 43.7%)(p < 0.05) (Fig 2C) were increased in Mdx+OVL mice compared to Mdx mice. To determine whether the activation of the calcineurin pathway needs to be involved to have the beneficial effects of OVL, we treated a group of Mdx+OVL mice with CsA (25 mg/kg), an inhibitor of this pathway (Fig 1). We have previously shown that CsA, at this administered dose, blocks the activation of the calcineurin pathway in murine dystrophic muscle [17]. Interestingly, we observed no difference (p > 0.05) between Mdx+OVL+CsA and Mdx+OVL mice concerning the force drop following lengthening contractions (Fig 2A), absolute maximal force (Fig 2B) and muscle weight (Fig 2C). However, CsA did alter the change in the rate of force development induced by OVL (p < 0.05)(Fig 2D), indicating that CsA does have an effect, since this functional parameter was not changed in Mdx+OVL+CsA, contrary

to Mdx+OVL. Together these results indicate that the beneficial effects of OVL on fragility and maximal force of the plantaris muscle were minimally affected by CsA in Mdx+OVL mice.

### A lower OVL also generated beneficial effects in Mdx mice (Experiment 2)

Next, we determined the effect of OVL on the lateral gastrocnemius (LG) muscle in *mdx* mice, a more physiological OVL than in the case of plantaris muscle (Experiment 2, Fig 1). The OVL of the LG muscle was less important that the OVL of the plantaris muscle because the amount of agonist muscle removed was lower in the case of the OVL of the LG muscle (the weight of the plantaris muscle is lower as compared to medial gastrocnemius muscle). In line with the lower OVL, the maximal force (Fig 3A) and weight (Fig 3B) gains in the OVL LG muscle were decreased compared to the OVL plantaris muscle (Fig 2B and 2C). Indeed, we found that LG muscle absolute maximal force (Fig 3A) ($p < 0.05$) and weight (Fig 3B) ($p < 0.001$) increased only 22.7% and 26.2%, respectively, in Mdx+OVL mice compared with Mdx mice. The force drops following lengthening contractions were reduced in the LG muscle from Mdx+OVL mice compared to Mdx mice. The remaining force following the 9th lengthening contraction was increased in the LG muscle from Mdx+OVL mice compared with Mdx mice (39.8% versus 25.5%) ($p < 0.01$) (Fig 3C), but to a lower extent in comparison to the OVL plantaris muscle (Fig 2A).

Together, these results show that improvements in fragility, maximal force and muscle weight are also achieved by the OVL of the LG muscle, although to a lesser extent than for plantaris muscle.

### OVL was also beneficial in the more severe dystrophic D2-mdx mice (Experiment 3)

We also studied the effect of OVL of the plantaris muscle in D2-*mdx* mice, a more severe murine DMD model (Experiment 3, Fig 1). We found that the force drops following lengthening contractions were reduced in D2-mdx+OVL mice compared with D2-mdx mice (Fig 4A). For example, the remaining force following the 9th lengthening contraction was increased in D2-mdx+OVL mice compared with D2-mdx mice ($p < 0.0001$)(66.1% versus 39.1%)(Fig 4A). In addition, absolute maximal force ($p < 0.01$)(Fig 4B) and muscle weight ($p < 0.0001$) (Fig 4C) increased by 48% and 49%, respectively, in D2-mdx+OVL mice compared with D2-mdx mice.

We also evaluated the protein expression of important modulators of muscle fragility and maximal force/hypertrophy. Using immunoblotting analysis, we found that OVL increased the protein levels of utrophin ($p < 0.05$) (Fig 5A and 5B) and cytoplamic γ-actin ($p < 0.05$) (Fig 5C and 5D) in plantaris muscle from D2-mdx+OVL, without affecting that of desmin (Fig 5E and 5F). The phosphorylation of Akt (p-Akt)($p < 0.05$) (Fig 6A and 6B) as well as the expression of MyoD ($p < 0.05$) (Fig 6C and 6D) were also increased in D2-mdx+OVL mice compared to D2-mdx mice. The amount of MHC-2a protein was not significantly changed by OVL (Fig 6C and 6E) neither the components of the oxidative respiratory chain (Fig 6F and 6G).

Together, these results indicate that the beneficial effects of OVL on muscle maximal force, weight and fragility were also observed in the plantaris muscle from D2-mdx+OVL mice, concomitantly with increased levels of different proteins known to play a role in fragility and maximal force/hypertrophy (utrophin, cytoplamic γ-actin, p-Akt and MyoD).

## Discussion

### OVL improved fragility in *mdx* mice treated or not with CsA and D2- *mdx* mice

The present study confirms our previous observations [20] showing that 1-month OVL alleviates the fragility, a major dystrophic functional feature, in a similar manner to dystrophin

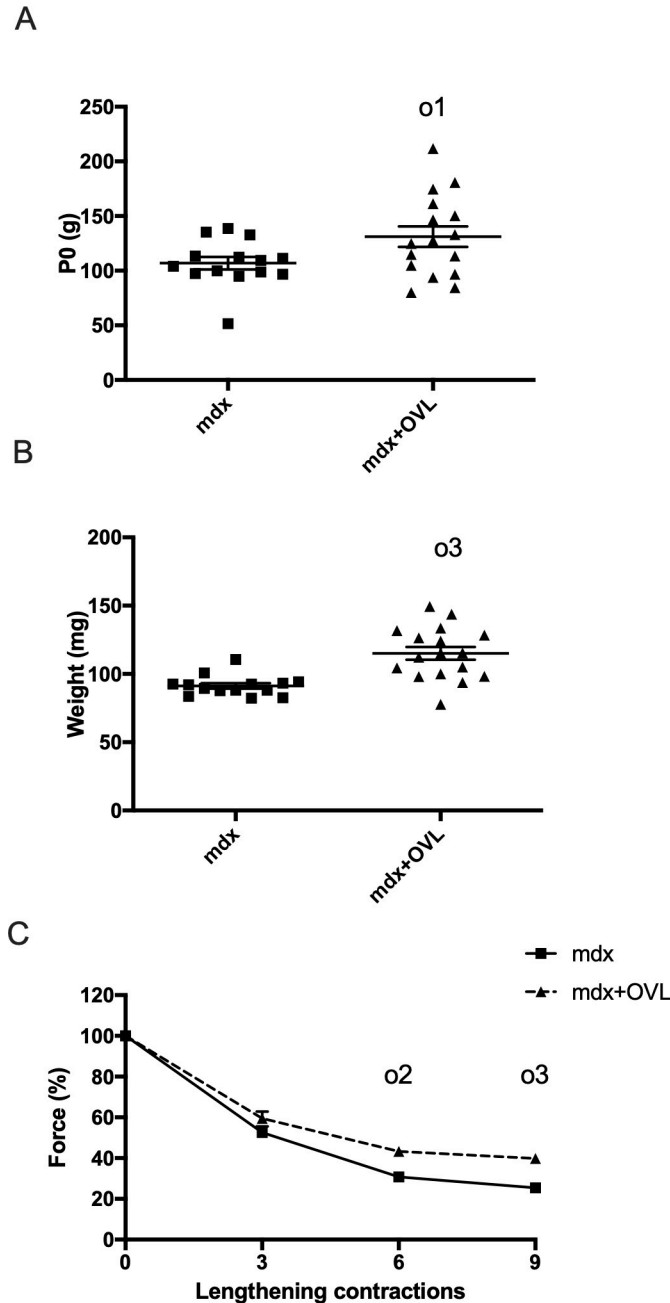

**Fig 3. Effect of OVL on lateral gastrocnemius muscle in Mdx mice.** (A) Absolute maximal force. n = 14–16 per group. (B) Muscle weight. n = 6–8 per group n = 14–17 per group. (C) Force drop following lengthening contractions (Fragility). n = 14–16 per group. Mdx+OVL: mechanically overloaded Mdx mice. Mdx: non-overloaded Mdx muscle. o1, o2, o3: significant different from Mdx ($p < 0.05$), ($p < 0.01$) and ($p < 0.001$) respectively.

restoration [31]. It was also recently reported an improvement in fragility after 6 training sessions comporting each 10 isometric maximal tetanic contractions per session in the *mdx* mice [19]. For the first time, we show that the beneficial effect of OVL was exerted in a muscle independent manner, independent of the degree of the OVL and the severity of the murine DMD model. This is very encouraging if we were to consider the prescription of resistance training by the physician for DMD patients since murine models are less affected than patients. Thus,

A

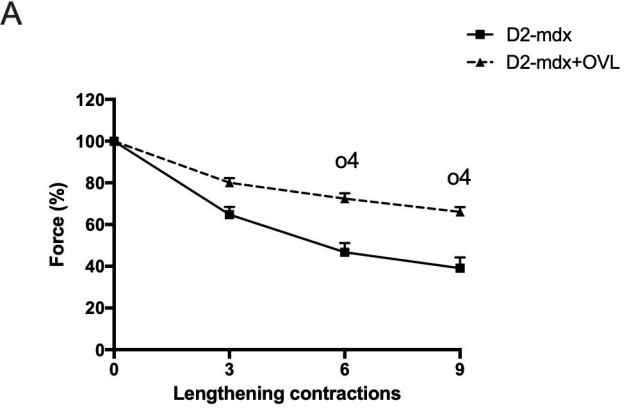

B

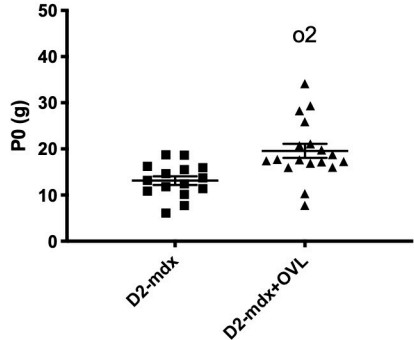

C

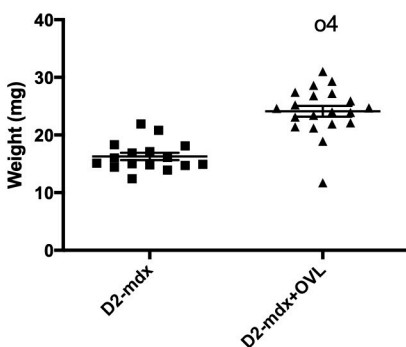

**Fig 4. Effect of OVL on plantaris muscle in D2-mdx mice: Physiological data.** (A) Force drop following lengthening contractions (Fragility). n = 14–16 per group. (B) Absolute maximal force. n = 15–18 per group. (C) Muscle weight. n = 6–8 per group. n = 16–20 per group. D2-mdx+OVL: mechanically overloaded D2-mdx mice. D2-mdx: non-overloaded D2-mdx muscle. o2, and o4: significant different from ($p < 0.01$) and ($p < 0.0001$) respectively.

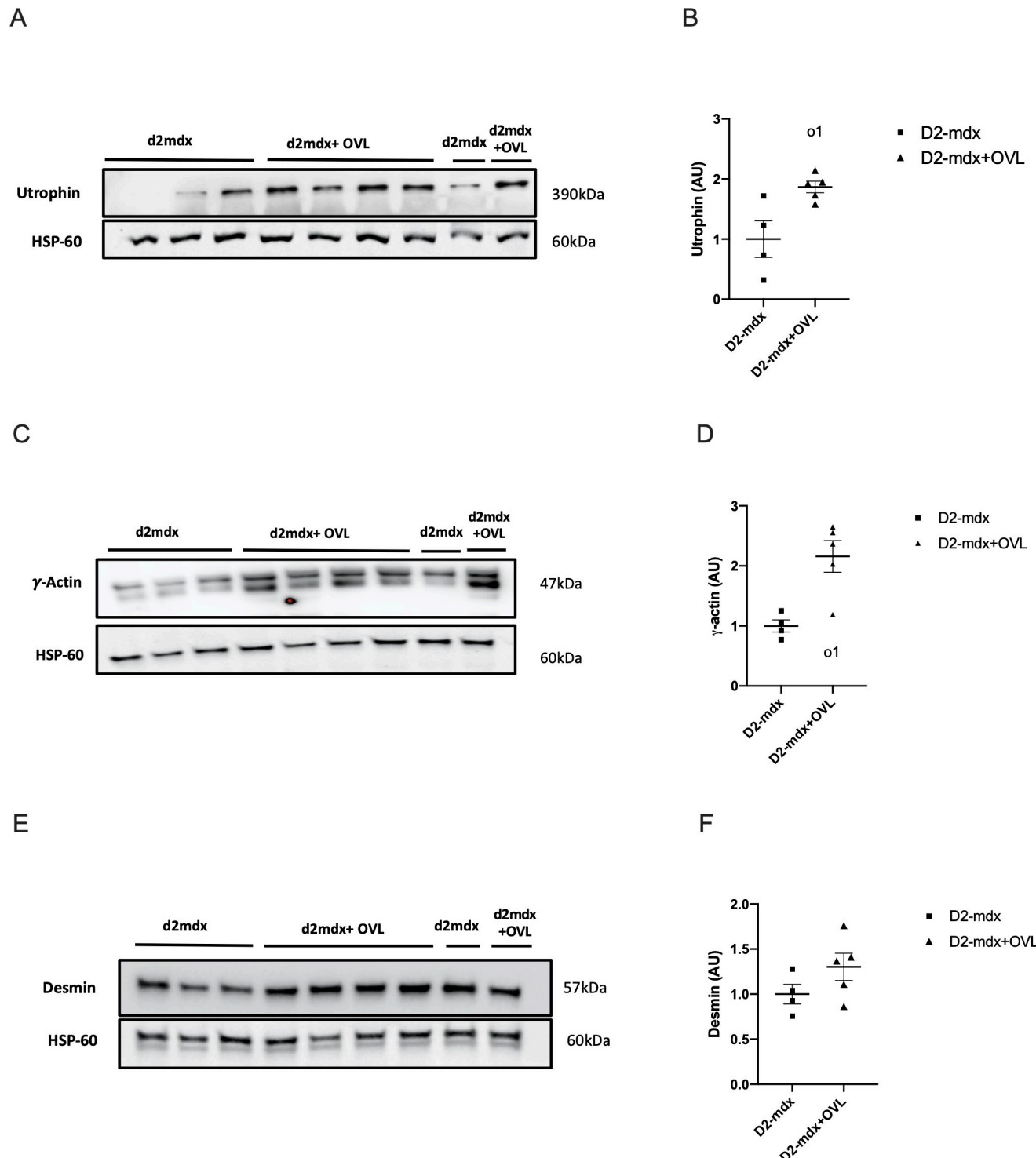

**Fig 5. Effect of OVL on plantaris muscle in D2-mdx mice: Expression of utrophin, cytoplamic γ-actin, and desmin.** (A) Representative image of a immunoblot showing utrophin band. (B) Utrophin protein levels. n = 4–5 per group. (C) Representative image of a immunoblot showing cytoplamic γ-actin band. (D) Cytoplamic γ-actin protein levels. n = 4–5 per group. (E) Representative image of a immunoblot showing desmin band. (F) Desmin protein levels. Immunoreactive bands were normalized to HSP-60 bands. Values are then expressed relative to mdx levels (mdx = 1). n = 4–5 per group. D2-mdx +OVL: mechanically overloaded D2-mdx mice. D2-mdx: non-overloaded D2-mdx muscle. o1: significant different from Mdx (p < 0.05).

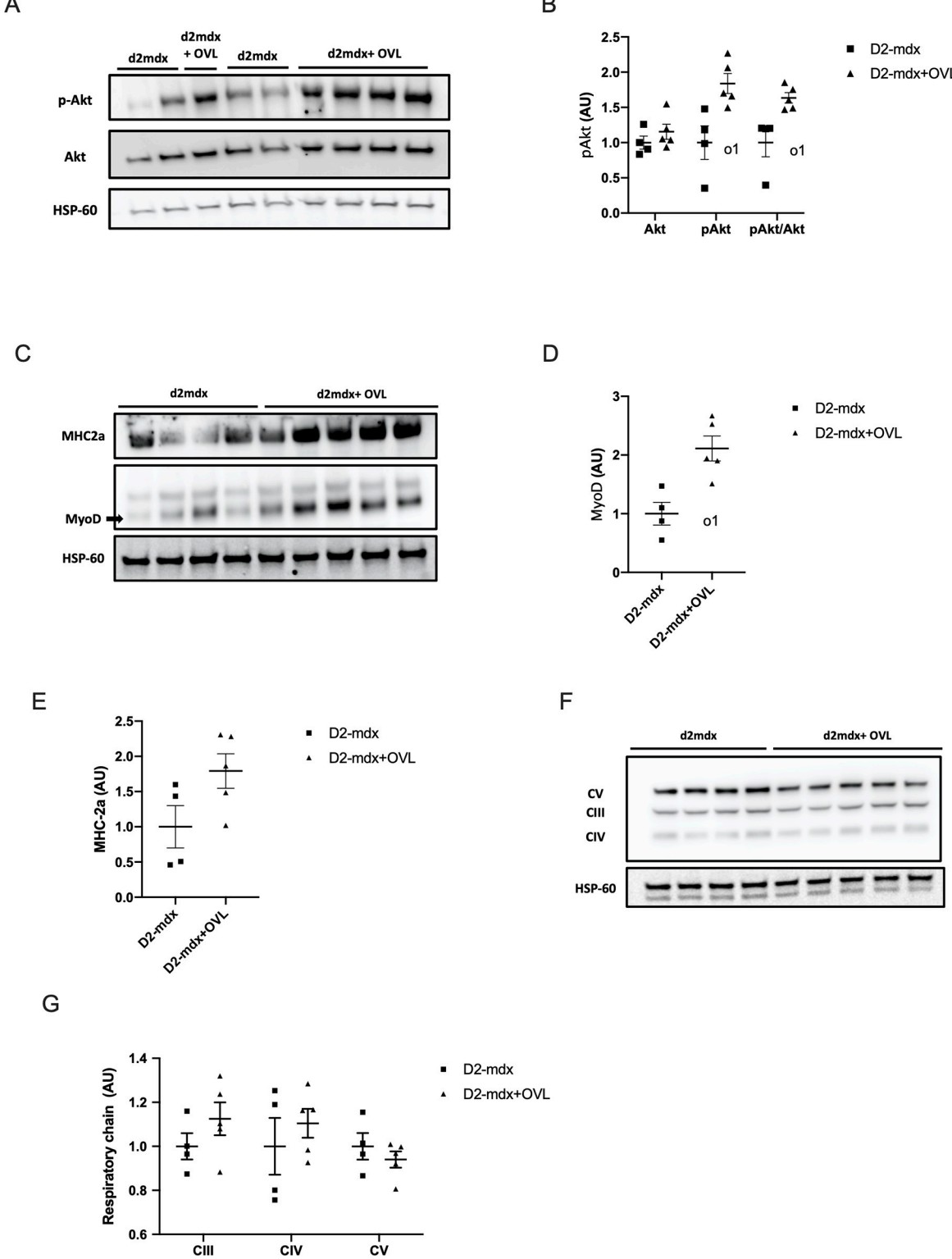

**Fig 6. Effect of OVL on plantaris muscle in D2-mdx mice: Expression of pAkt, MyoD, MHC-2a and the complexes of the respiratory chain.** (A) Representative image of a immunoblot showing the bands of Akt and p-Akt. (B) p-Akt levels. n = 4–5 per group. (C) Representative image of a immunoblot showing the bands of MyoD and MHC-2a. (D) MyoD protein levels. n = 4–5 per group. (E) MHC-2a protein levels. n = 4–5 per group. (F) Representative image of a immunoblot showing the bands of the complexes (CIII, CIV and CV) of the respiratory chain. (G) Protein levels of the complexes (CIII, CIV and CV) of the respiratory chain protein levels. n = 4–5 per

group. Immunoreactive bands were normalized to HSP-60 bands. Values are then expressed relative to mdx levels (mdx = 1). D2-mdx +OVL: mechanically overloaded D2-mdx mice. D2-mdx: non-overloaded D2-mdx muscle. o1: significant different from Mdx (p < 0.05).

our results suggest that the reduction of fragility can contribute to the improvement of the quality of life of the DMD patients insofar as they would suffer less this loss of function following lengthening contractions, together with having an increased absolute maximal force (see below).

Interestingly, we found that the calcineurin inhibitor CsA did not block the OVL-induced decrease in fragility in *mdx* mice, suggesting that activation of the calcineurin pathway is not the mechanism involved in this beneficial effect of OVL in murine dystrophic mice. It is possible that the calcineurin pathway was not activated by OVL. One study reports the absence of activation of the calcineurin pathway by OVL in healthy rodents [11], but several other do not [23, 24, 32]. One limitation of our study is that we do not provide a direct measure of calcineurin pathway activation. This deserves to be studied more closely in the future. Whatever, the present and previous results [17] suggest that different types of chronic exercise (voluntary running and OVL) exert the same improvement of fragility in murine dystrophic muscle via different mechanisms, since the beneficial effect of the two different types of chronic muscular exercise was blocked by CsA [17] or not (the present study).

Moreover, the improvement in fragility by OVL does not seem to be explained by an increase in the number of potentially less fragile muscle fibers [1] since our results suggest that the percentage of slower and more oxidative fibers remains unchanged in the D2-*mdx* (no increase in the amounts of MHC-2a and complexes of the respiratory chain), in contrast to what was observed in *mdx* mice [22]. However, it would be interesting to confirm the absence of muscle fiber type transition using a histological study and a larger sample size. We also eliminated the possibility that a decrease in MyoD expression in murine dystrophic mice would contribute to the improvement in fragility. It has recently been suggested that blocking the induction of the MyoD-dependent fetal gene program in myofibers improves sarcolemmal membrane stability and fragility [10]. However, we reported the contrary, an increase in the expression of MyoD in D2-mdx+OVL mice.

Based on previous studies [6, 8, 33], we next formulated the hypothesis that the improved fragility in murine dystrophic mice was associated to hypertrophy (increased muscle weight) and/or maximal force gain for the following reasons. First, we found that the improvement in fragility was roughly proportional to the hypertrophy and maximal force gain in *mdx* mice (the improvement in fragility was greater in the OVL plantaris compared to OVL LG gastrocnemius muscles). Consistently, there were correlations between fragility and muscle weight (S1 Table). Second, we found an increase in the Akt phosphorylation in response to OVL in D2-*mdx* mice, as previously shown in *mdx* mice [22], suggesting the activation of the Akt pathway. The activation of the Akt/mTOR pathway by OVL is indispensable to hypertrophy (and likely maximal force gain) [11]. Interestingly, we found a correlation between the activation of the Akt/mTOR pathway and hypertrophy in D2-*mdx* mice (S1 Table). Third, genetic activation of Akt in *mdx* mice leads to hypertrophy and increased maximal force as well as reduced fragility [6]. Fourth, genetic inactivation of mTOR in healthy muscle results in atrophy, reduced maximal force and increased fragility [8]. In a previous study, we found that OVL increased p-S6 in *mdx* mice, suggesting the activation of the mTOR pathway [22]. Fifth, the administration of formoterol in *mdx* mice, that induces hypertrophy and greater maximal force, decreases the fragility [33]. Formoterol is reported to activate the Akt pathway [13]. Sixth, inactivity, achieved by leg immobilization, reduced maximal force and increased fragility in *mdx* mice [16]. Together these results suggest that there is a link between the improved

fragility in OVL dystrophic mice and maximal force and weight gains and the activation of the Akt and mTOR pathways, even if the downstream events have not yet been identified [6]. However, this link remains to be demonstrated in the future. It has also been reported that the Akt and mTOR pathways control the expression of several proteins providing a physical link between the extracellular matrix and the intracellular cytoskeleton, such as dystrophin, utrophin and desmin [6, 8]. These proteins are also known to modulate fragility in dystrophic muscle [2, 3, 5, 34]. However, we demonstrate that utrophin was increased by OVL in D2-*mdx* mice but not in *mdx* mice [22] and desmin was not increased by OVL in D2-*mdx* mice. It is possible however that the increased cytoplamic γ-actin in D2-mdx+OVL mice contributed to the improved fragility since transgenic overexpression of cytoplamic γ-actin reduces fragility in *mdx* mice [4]. Finally, and in a manner always linked to hypertrophy, it is possible that the reduction in fragility in response to OVL is due to the increase in the ratio of fiber length to muscle length [35]. This increase would have the effect of reducing the percentage of lengthening at the fiber level, since we achieved a stretch of 10% of the length of the muscle.

## OVL increased maximal force production in both mild and severe murine DMD models

In addition to the improvement in fragility, OVL markedly increased the absolute maximal force of the *mdx* mice, independent the lower leg muscle concerned (plantaris or LG gastrocnemius muscles), in line with previous results [20, 22]. A smaller improvement of approximately 12% in absolute maximal force after resistance training was also reported in *mdx* mice [19]. In the future, it would be interesting to know if the restoration of these parameters is complete (no more difference with healthy muscle). What is new is that this maximal force gain was also observed in the case of a severe murine model of DMD (D2-*mdx* mice), closer to the human DMD phenotype. Thus, unlike the absence of desmin [36], another protein of the costamer, the loss of dystrophin does not prevent the increase in maximal force induced by OVL. This beneficial effect of OVL is related to hypertrophy (increased muscle weight), increased level of p-Akt and MyoD in D2-*mdx* (the present study) and *mdx* mice [22], increased level of p-S6 in *mdx* mice [22], thus possibly the activation of the Akt/mTOR pathway that is indispensable for the OVL-induced hypertrophy [11]. Moreover, we found that the hypertrophy was not blocked by CsA, suggesting that the calcineurin pathway does not play an important role which in line with a previous study [11] but not for unknown reasons with others studies [23, 24, 32], which have been widely debated [37]. This increase in maximal force in D2-*mdx* mice after 1-month of OVL was interesting because it has been previously reported that the restoration of dystrophin in response to a gene therapy approach in D2-*mdx* mice was able or not to increase maximal force [38, 39]. Of note, several other types of fragility-improving preclinical therapies do not increase absolute maximal force in mouse models of DMD, i.e., could not potentially mitigate muscle weakness, and sometimes even increased it [9, 18, 27, 40, 41]. Finally, another aspect of muscular performance that is improved by OVL is the ability to produce force rapidly. We show that the increase in the rate of force development was diminished by CsA, suggesting that fiber type is not the only factor determining this ability [29].

## Conclusion

Our results show that OVL reduced the susceptibility to contraction-induced force drop, independent of the severity of the OVL or the type of leg muscle, in both mild and severe murine DMD models. This beneficial effect of OVL on muscle fragility is associated to maximal force gain, hypertrophy, increased expression of p-Akt and cytoplamic γ-actin, and was not

inhibited by CsA which is known to block the activation of the calcineurin pathway. Moreover, OVL also induced substantial increase in maximal force related to muscle growth. Thereby, this murine model of resistance training improved two important functional dystrophic features in both mildly and severely affected mice. Collectively, studies using more or less severe murine models of DMD [19, 20, 22] as well as the study in DMD patients [42] support the idea that it would be interesting to determine that resistance training, comprising muscle contractions in small numbers but developing high levels of force, would benefit DMD and Becker patients by allowing notable strength gains.

## Supporting information

**S1 Table. Correlations.**
(DOCX)

**S1 File. Minimal data.** Plantaris mdx, Force (experiment 1).
(PDF)

**S2 File. Minimal data.** Lat Gast Mdx, force (experiment 2).
(PDF)

**S3 File. Minimal data.** Plantaris D2-mdx, force (experiment 3).
(PDF)

**S4 File. Minimal data.** Plan D2-mdx, Blots values (experiment 3).
(PDF)

**S1 Raw image. Fig 5A UTROPHIN.** Image of blot.
(PDF)

**S2 Raw image. Fig 5C. GAMA ACTIN.** Image of blot.
(PDF)

**S3 Raw image. Fig 5E. DESMIN.** Image of blot.
(PDF)

**S4 Raw image. Fig 6A. P-AKT.** Image of blot.
(PDF)

**S5 Raw image. Fig 6C. MyoD and MHC2a.** Image of blot.
(PDF)

**S6 Raw image. Fig 6F. Respiratory Chain.** Image of blot.
(PDF)

## Acknowledgments

We thank Gillian Butler-Browne for her invaluable help.

## Author Contributions

**Conceptualization:** Onnik Agbulut, Arnaud Ferry.

**Formal analysis:** Arnaud Ferry.

**Funding acquisition:** Denis Furling, Onnik Agbulut, Arnaud Ferry.

**Investigation:** Medhi Hassani, Dylan Moutachi, Mégane Lemaitre, Alexis Boulinguiez, Arnaud Ferry.

**Methodology:** Onnik Agbulut, Arnaud Ferry.

**Project administration:** Denis Furling.

**Writing – original draft:** Medhi Hassani, Arnaud Ferry.

**Writing – review & editing:** Alexis Boulinguiez, Onnik Agbulut.

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
