## [Decision Letter · Decision Letter 0]

3 Sep 2023

PONE-D-23-21038Beneficial effects of resistance training on both mild and severe mouse dystrophic muscle function as a preclinical option for Duchenne muscular dystrophyPLOS ONE

Dear Dr. Ferry,

Thank you for submitting your manuscript to PLOS ONE. After careful consideration, we feel that it has merit but does not fully meet PLOS ONE’s publication criteria as it currently stands. Therefore, we invite you to submit a revised version of the manuscript that addresses the points raised during the review process. The referees all found that the influence of exercise and muscle loading in DMD is of importance to the field. However, there are concerns that need to be addressed. Improved description of the experimental design, addressing the methods concerns, comparison with other studies and directly testing for significant relationships between one of the parameters such as hypertrophy and force loss or pAKT levels and muscle mass is needed.

We look forward to receiving your revised manuscript.

Kind regards,

Aldrin V. Gomes, Ph.D.

Academic Editor

PLOS ONE

Journal Requirements:

4. We note that Figure 1 in your submission contain copyrighted images. All PLOS content is published under the Creative Commons Attribution License (CC BY 4.0), which means that the manuscript, images, and Supporting Information files will be freely available online, and any third party is permitted to access, download, copy, distribute, and use these materials in any way, even commercially, with proper attribution. For more information, see our copyright guidelines: http://journals.plos.org/plosone/s/licenses-and-copyright.

Reviewers' comments:

Reviewer's Responses to Questions

**Comments to the Author**

1. Is the manuscript technically sound, and do the data support the conclusions?

Reviewer #1: Partly

Reviewer #2: No

Reviewer #3: Yes

2. Has the statistical analysis been performed appropriately and rigorously? 

Reviewer #1: Yes

Reviewer #2: Yes

Reviewer #3: Yes

3. Have the authors made all data underlying the findings in their manuscript fully available?

Reviewer #1: Yes

Reviewer #2: Yes

Reviewer #3: Yes

4. Is the manuscript presented in an intelligible fashion and written in standard English?

Reviewer #1: Yes

Reviewer #2: Yes

Reviewer #3: Yes

5. Review Comments to the Author

Reviewer #1: The manuscript by Hassani et. al. investigates the beneficial impact of muscle activity in mouse models of muscular dystrophy. Specifically, the authors show that synergistic ablation induces muscle hypertrophy, maximal force production, and resistance to eccentric damage. While demonstrated previously, this has been expanded to be present even with lower levels of hypertrophy, in a more severely affected mouse model and to not be dependent on calcineurin signaling. The manuscript is generally well written. The influence of exercise and muscle loading in DMD is of importance to the field with questions that this paper can help to address. However, there are substantial concerns that dampen enthusiasm listed below.

The decrease fragility could simply be more sarcomeres in series leading to less sarcomeric strain or strain rate with the eccentric contraction. The description of how lengths are recorded is incomplete. It appears Lo was set to where maximum force was achieved, but how was Lo then measured in situ? Where calipers used and was any tendon length included? This is important given that the lengthening contractions are set to 10% of Lo and Lo is expected to change with overload.

Although it did not turn out to induce major effects the treatment with CsA is thought to act through inhibiting slow fiber programs through Calcineurin. However, only one myosin heavy chain molecule was examined. It would be informative to look at fiber types more directly to see if that was related at all to the susceptibility to damage as postulated.

The experimental design section is helpful to show what was done. However, the rationale for some decisions could be more fully explained. Particularly, it may be helpful to include wildtype animals so the degree of recovery could be stated for certain parameters. Also, it isn’t clear why a bilateral design is not used to compare the contralateral limb to that of the OVL.

There are a number of associations that are proposed between parameters studied, such as that between AKT and muscle growth. Given that many of the parameters are investigated within the same muscle it would be helpful to directly test for significant relationships between parameters such as hypertrophy and force loss or pAKT levels and muscle mass.

Minor comments:

Methods: It is stated that a velocity of 5.5 mm/s was used that is equivalent to a 0.85 fiber length/s velocity. This would assume a constant fiber length of ~6.5 mm. How is this determined, especially in light of potentially changing Lo?

Methods: The methods state that lengthening contractions were induced in the TA, however that is not consistent with the rest of the manuscript.

Fig 2C: Weight is misspelled on the axis.

Fig 2D: The rationale for separating out the different phases of force development is not clear. It may be easier to just show one value (perhaps 0-50%).

Fig 2D: The rate of force development is slowed with CsA, but this would seem counter to expectations of CaN promoting slower fibers (thus blocking it would promote faster fibers). A comment in the discussion could be warranted.

Fig 3C: It isn’t clear if error bars are present in this figure.

Fig 5: The lane labels are not very clear.

Discussion: The statement that improvement in fragility cannot be explained by an increase in less fragile fibers could e overly strong. While a significant change in MHC2a was not observed there is a shift in the mean and with low sample sizes here this appears under powered.

Discussion: Research is cited that gene therapy does not increase muscle force in the D2 model. However, there is data showing substantial increases in muscle force with microdystrophin gene therapy (Cernisova et. al. Int J Mol Sci 2023).

Reviewer #2: The study describes the benefits of resistance training in two mouse DMD models, offering preclinical data that might be relevant to human disease. Results are solid and well presented, yet the study is purely descriptive and the manuscript needs revision before it might be acceptable for publication.

Main concerns with this study are that it offers little mechanistic insight and no direct link to applicability to human medicine. The title should be improved to better phrase the study.

The underlying mechanisms of benefit have not been investigated in depth. Administering CsA leads the authors to conclude that calcineurin signalling is not involved, and it is speculated that Akt/mTor might be involved. However, no specific data is offered that might corroborate this statement, and it therefore needs to be downplayed. The study appears to me somewhat immature., and authors need to clearly acknowledge limitations and future routes forward. Lines 344-370 need to be reduced in fit with the shortcomings of the study.

The manuscript would benefit from comparison with other studies given in a comprehensive table. Also, relevance to human medicine should be better explored. What could be a recommended (improved) training strategy based upon the study’s findings? What is the added value for the clinic? Are there any significant differences between mdx and D2-mdx that aid in comparison of severity? Becker MD is not mentioned anywhere, give this some thought and add to the discussion. In view of human DMD being severe, how can resistance training be implement in the window of ambulation? Authors should more thoroughly discuss the translational value of the study.

There are several method concerns. Describe how the CsA dose and administration was determined. Was this based on literature or on own preliminary tests? Give concentrations of antibodies used for western blots in µg/ml IgG and not as a dilution factor. Why was gamma-actin chosen as a loading control, while most use alpha- or beta-? Also, the use of HSP60 as a mitochondrial marker needs to be explained (as well as its preference to VDAC). Legends to fig 5 and 6 do not mention HSP60, nor if the graphs of protein levels are ratio’s to HSP60. Explain rationale of normalization to mitochondrial content. Full western blots need to be provided as a supplement. Figure legends need to specify precise numbers of mice per group, and not ranges.

Minor remarks:

Line 280 benefic

Line 293 cytoplamic

Reviewer #3: I think this research paper is very detailed and thorough. It is technically sound, the data supports the conclusions and the statistical analysis is very rigorous. It just needs some small revisions.

I have one main area of confusion. For the first sub -header of the discussion section, I was very confused by the wording. "OVL improved fragility in both mdx and D2-mdx mice, treated or not with CsA" makes it sound like both the mdx and the D2-mdx mice were treated with CsA. I had to go back to the methods section to double-check that the D2-mdx mice were never treated with CsA.

In addition, I was wondering why you all did not include a CsA group in the D2-mdx mice?

Lastly, I believe almost every time the cytoplasmic gamma-actin are mentioned in the paper, cytoplasmic is spelled incorrectly. I saw "cytoplamic" multiple times throughout the paper.

6. PLOS authors have the option to publish the peer review history of their article (what does this mean?). If published, this will include your full peer review and any attached files.

Reviewer #1: No

Reviewer #2: No

Reviewer #3: No

---

## [Author Response · Author response to Decision Letter 0]

7 Nov 2023

Academic Editor:

The referees all found that the influence of exercise and muscle loading in DMD is of importance to the field. However, there are concerns that need to be addressed. Improved description of the experimental design, addressing the methods concerns, comparison with other studies and directly testing for significant relationships between one of the parameters such as hypertrophy and force loss or pAKT levels and muscle mass is needed. AUTHORS' RESPONSE: This was revised.

Journal Requirements:

Upon re-submitting your revised manuscript, please upload your study’s minimal underlying data set as either Supporting Information files or to a stable, public repository and include the relevant URLs, DOIs, or accession numbers within your revised cover letter. For a list of acceptable repositories, please see http://journals.plos.org/plosone/s/data-availability#loc-recommended-repositories. Any potentially identifying patient information must be fully anonymized. AUTHORS' RESPONSE: Supporting information files were uploaded.

3. PLOS ONE now requires that authors provide the original uncropped and unadjusted images underlying all blot or gel results reported in a submission’s figures or Supporting Information files. This policy and the journal’s other requirements for blot/gel reporting and figure preparation are described in detail athttps://journals.plos.org/plosone/s/figures#loc-blot-and-gel-reporting-requirements and https://journals.plos.org/plosone/s/figures#loc-preparing-figures-from-image-files. When you submit your revised manuscript, please ensure that your figures adhere fully to these guidelines and provide the original underlying images for all blot or gel data reported in your submission. See the following link for instructions on providing the original image data: https://journals.plos.org/plosone/s/figures#loc-original-images-for-blots-and-gels. AUTHORS' RESPONSE: Supporting information files (images of the blots) were uploaded.

In your cover letter, please note whether your blot/gel image data are in Supporting Information or posted at a public data repository, provide the repository URL if relevant, and provide specific details as to which raw blot/gel images, if any, are not available. Email us at plosone@plos.org if you have any questions. AUTHORS' RESPONSE: This was done.

4. We note that Figure 1 in your submission contain copyrighted images. All PLOS content is published under the Creative Commons Attribution License (CC BY 4.0), which means that the manuscript, images, and Supporting Information files will be freely available online, and any third party is permitted to access, download, copy, distribute, and use these materials in any way, even commercially, with proper attribution. For more information, see our copyright guidelines: http://journals.plos.org/plosone/s/licenses-and-copyright.

Please upload the completed Content Permission Form or other proof of granted permissions as an ""Other"" file with your submission. AUTHORS' RESPONSE: This was done, we upload the permission.

Reviewers' comments:

Reviewer #1: 

The decrease fragility could simply be more sarcomeres in series leading to less sarcomeric strain or strain rate with the eccentric contraction. The description of how lengths are recorded is incomplete. It appears Lo was set to where maximum force was achieved, but how was Lo then measured in situ? Where calipers used and was any tendon length included? This is important given that the lengthening contractions are set to 10% of Lo and Lo is expected to change with overload. AUTHORS' RESPONSE: We thank the reviewer for these excellent remarks. It is possible that the longitudinal growth of the fibers is responsible for the improvement in fragility. We now discussed the possible increase in the ratio of fiber length to muscle length in response to OVL (Jorgenson et al 2020) could be the cause of the improvement in fragility (Discussion). Muscle length (L0) was measured from proximal extremity (knee) to myotendinous junction (distal extremity) using caliper. Distal tendon length was not included. These informations are now included (Materials and Methods).

Although it did not turn out to induce major effects the treatment with CsA is thought to act through inhibiting slow fiber programs through Calcineurin. However, only one myosin heavy chain molecule was examined. It would be informative to look at fiber types more directly to see if that was related at all to the susceptibility to damage as postulated. AUTHORS' RESPONSE: In two previous studies (Ferry et al 2015, Joanne et al 2012), we did not observe any change in the expression of MHC-2b in response to OVL in the plantaris muscle of mdx mice. Additionally, the change in MHC-1 expression was modest. Unfortunately, we did not prepare muscle samples for histological study. Thus, we revised the statement that improvement in fragility cannot be explained by an increase in less fragile fibers. Moreover, we added that it would be interesting to confirm the absence of muscle fiber type transition using a histological study and a larger sample size (Discussion).

The experimental design section is helpful to show what was done. However, the rationale for some decisions could be more fully explained. Particularly, it may be helpful to include wildtype animals so the degree of recovery could be stated for certain parameters. Also, it isn’t clear why a bilateral design is not used to compare the contralateral limb to that of the OVL. AUTHORS' RESPONSE: We agree, it would have been interesting to study WT mice to determine the degree of recovery compared to WT mice. This idea has been added (Discussion). A bilateral experimental design would indeed be more interesting, but we were always afraid that the mouse would not use much of the leg that was operated on. It is difficult/complicated to verify this hypothesis. Therefore, we followed the protocol of Roy's study in 1995.

There are a number of associations that are proposed between parameters studied, such as that between AKT and muscle growth. Given that many of the parameters are investigated within the same muscle it would be helpful to directly test for significant relationships between parameters such as hypertrophy and force loss or pAKT levels and muscle mass. AUTHORS' RESPONSE: We thank you for this remark. We studied these relationships in the revised manuscript (Discussion, S1 Table). We found correlations between fragility (the percentage of force remaining following 9 lengthening contractions) and the muscle weight in the 3 experiments. There were also correlations between the muscle weight and the pAkt/Akt ratio. 

Minor comments:

Methods: It is stated that a velocity of 5.5 mm/s was used that is equivalent to a 0.85 fiber length/s velocity. This would assume a constant fiber length of ~6.5 mm. How is this determined, especially in light of potentially changing Lo? AUTHORS' RESPONSE: This was revised. The velocity was kept constant.

Methods: The methods state that lengthening contractions were induced in the TA, however that is not consistent with the rest of the manuscript. AUTHORS' RESPONSE: This was revised.

Fig 2C: Weight is misspelled on the axis. AUTHORS' RESPONSE: This was revised.

Fig 2D: The rationale for separating out the different phases of force development is not clear. It may be easier to just show one value (perhaps 0-50%). AUTHORS' RESPONSE: The mechanisms responsible for RFD vary over time (Rodrıguez-Rosell et al 2018). This information has been added (Materials and Methods).

Fig 2D: The rate of force development is slowed with CsA, but this would seem counter to expectations of CaN promoting slower fibers (thus blocking it would promote faster fibers). A comment in the discussion could be warranted. AUTHORS' RESPONSE: This was done (Discussion).

Fig 3C: It isn’t clear if error bars are present in this figure. AUTHORS' RESPONSE: SEM are present (the error bars were shorter than the size of the symbol).

Fig 5: The lane labels are not very clear. AUTHORS' RESPONSE: This was revised.

Discussion: The statement that improvement in fragility cannot be explained by an increase in less fragile fibers could e overly strong. While a significant change in MHC2a was not observed there is a shift in the mean and with low sample sizes here this appears under powered. AUTHORS' RESPONSE: This statement was revised (Discussion).

Discussion: Research is cited that gene therapy does not increase muscle force in the D2 model. However, there is data showing substantial increases in muscle force with microdystrophin gene therapy (Cernisova et. al. Int J Mol Sci 2023). AUTHORS' RESPONSE:This was revised (Discussion).

*********

Reviewer #2: 

Main concerns with this study are that it offers little mechanistic insight and no direct link to applicability to human medicine. The title should be improved to better phrase the study.

The underlying mechanisms of benefit have not been investigated in depth. Administering CsA leads the authors to conclude that calcineurin signalling is not involved, and it is speculated that Akt/mTor might be involved. However, no specific data is offered that might corroborate this statement, and it therefore needs to be downplayed. The study appears to me somewhat immature., and authors need to clearly acknowledge limitations and future routes forward. Lines 344-370 need to be reduced in fit with the shortcomings of the study. AUTHORS' RESPONSE: We now clearly acknowledge limitations and future routes forward (Discussion). We found that Akt phosphorylation was increased by OVL. In previous studies (Bodine et al 2001, Blaauw et al 2008), Akt was demonstrated using antibodies specific for phospho-Akt, with Immunoblot analysis of protein extracts from muscle. Some data were also added to support the hypothesis that Akt/mTOR, hypertrophy and force gain might be associated (S1 Table), as suggested by Reviewer #1. We found correlations between fragility and hypertrophy, and hypertrophy and Akt/mTOR activation. In addition, reviewer #1's remarks led us to suggest adding a very interesting explanation related to hypertrophy. Indeed, it is possible that the improvement in fragility could be linked to the longitudinal growth of the fibers, this phenomenon also participating in hypertrophy (Discussion).

The manuscript would benefit from comparison with other studies given in a comprehensive table. AUTHORS' RESPONSE: Only 2 studies (ours) studied the effect of OVL on fragility and maximal force (Joanne et al 2012, Ferry et al 2015). A 3rd study, that of Lindsay et al (2019), determined the effects of isometric resistance training on maximal force and fragility. They reported that OVL and isometric resistance training improve maximal force and fragility in mdx mice (plantaris and EDL muscles). Comparisons are now made (Discussion).

Also, relevance to human medicine should be better explored. What could be a recommended (improved) training strategy based upon the study’s findings? What is the added value for the clinic? Are there any significant differences between mdx and D2-mdx that aid in comparison of severity? Becker MD is not mentioned anywhere, give this some thought and add to the discussion. In view of human DMD being severe, how can resistance training be implement in the window of ambulation? Authors should more thoroughly discuss the translational value of the study. AUTHORS' RESPONSE: We did not find any notable difference between mdx and D2-mdx mice, which is encouraging for the application of this type of training to DMD patients. This idea was already stated in the Discussion. We more thoroughly discuss the translational value of the study, as suggested (Conclusion).

There are several method concerns. Describe how the CsA dose and administration was determined. Was this based on literature or on own preliminary tests? AUTHORS' RESPONSE: The dose of CsA and administration were based on previous studies and our own experience, as already stated in Materials and Method.

Give concentrations of antibodies used for western blots in µg/ml IgG and not as a dilution factor. AUTHORS' RESPONSE: Indeed it is not relevant to mention dilution. This has been fixed.

Why was gamma-actin chosen as a loading control, while most use alpha- or beta-? Also, the use of HSP60 as a mitochondrial marker needs to be explained (as well as its preference to VDAC). Legends to fig 5 and 6 do not mention HSP60, nor if the graphs of protein levels are ratio’s to HSP60. Explain rationale of normalization to mitochondrial content. Full western blots need to be provided as a supplement. AUTHORS' RESPONSE: Gamma-actin was not chosen as a loading control, unlike HSP60. Gamma-actin is one of the proteins of interest, studied among several others. It is correct that protein levels are ratio's to HSP60 and the expressed relative to mdx. This is now mentioned in the revised manuscript (methods and legend figures). The level of HSP60 did not vary between groups.

Figure legends need to specify precise numbers of mice per group, and not ranges. AUTHORS' RESPONSE: It is possible to know precisely the number of mice in each group by examining the figures (individual values are shown) and supporting information in the revised manuscript.

Minor remarks:

Line 280 benefic AUTHORS' RESPONSE: This has been corrected in the new version of the manuscript.

Line 293 cytoplamic AUTHORS' RESPONSE: This has been corrected in the new version of the manuscript.

Reviewer #3: 

I have one main area of confusion. For the first sub -header of the discussion section, I was very confused by the wording. "OVL improved fragility in both mdx and D2-mdx mice, treated or not with CsA" makes it sound like both the mdx and the D2-mdx mice were treated with CsA. I had to go back to the methods section to double-check that the D2-mdx mice were never treated with CsA. AUTHORS' RESPONSE: This has been corrected.

In addition, I was wondering why you all did not include a CsA group in the D2-mdx mice? AUTHORS' RESPONSE: As we did not find an effect of CsA in the mdx mice, we had no reason to think that this might be different in the D2-mdx mouse. For the OVL effects that we compared, we did not find any difference overall between the 2 murine DMD models.

Lastly, I believe almost every time the cytoplasmic gamma-actin are mentioned in the paper, cytoplasmic is spelled incorrectly. I saw "cytoplamic" multiple times throughout the paper. AUTHORS' RESPONSE: This has been corrected.

---

## [Decision Letter · Decision Letter 1]

28 Nov 2023

Beneficial effects of resistance training on both mild and severe mouse dystrophic muscle function as a preclinical option for Duchenne muscular dystrophy

PONE-D-23-21038R1

Dear Dr. Ferry,

We’re pleased to inform you that your manuscript has been judged scientifically suitable for publication and will be formally accepted for publication once it meets all outstanding technical requirements.

Kind regards,

Aldrin V. Gomes, Ph.D.

Academic Editor

PLOS ONE

Additional Editor Comments (optional):

Reviewers' comments:

Reviewer's Responses to Questions

**Comments to the Author**

1. If the authors have adequately addressed your comments raised in a previous round of review and you feel that this manuscript is now acceptable for publication, you may indicate that here to bypass the “Comments to the Author” section, enter your conflict of interest statement in the “Confidential to Editor” section, and submit your "Accept" recommendation.

Reviewer #1: (No Response)

Reviewer #3: All comments have been addressed

2. Is the manuscript technically sound, and do the data support the conclusions?

Reviewer #1: Partly

Reviewer #3: Yes

3. Has the statistical analysis been performed appropriately and rigorously? 

Reviewer #1: Yes

Reviewer #3: Yes

4. Have the authors made all data underlying the findings in their manuscript fully available?

Reviewer #1: Yes

Reviewer #3: Yes

5. Is the manuscript presented in an intelligible fashion and written in standard English?

Reviewer #1: Yes

Reviewer #3: Yes

6. Review Comments to the Author

Reviewer #1: The revised manuscript by Hassani et. al. makes notable improvements in the investigation of muscle overload in dystrophic muscles. The added relationships between hypertrophy and eccentric damage and pAKT support the conclusions. Importantly the limitations of the study have been more thoroughly addressed by the authors which prevent more strong conclusions from being drawn. The topic of overloading dystrophic muscle is of interest and the result of decreased susceptibility to damage is clear although the mechanistic insight is limited.

Reviewer #3: The author addressed all comments and concerns made to the manuscript. The manuscript is sound, the data supports the conclusions, and the statistical analysis has been performed appropriately and rigorously.

7. PLOS authors have the option to publish the peer review history of their article (what does this mean?). If published, this will include your full peer review and any attached files.

Reviewer #1: No

Reviewer #3: No

---

## [Editor Report · Acceptance letter]

4 Dec 2023

PONE-D-23-21038R1 

Beneficial effects of resistance training on both mild and severe mouse dystrophic muscle function as a preclinical option for Duchenne muscular dystrophy 

Dear Dr. Ferry:

I'm pleased to inform you that your manuscript has been deemed suitable for publication in PLOS ONE. Congratulations! Your manuscript is now with our production department. 

Kind regards, 

on behalf of

Dr. Aldrin V. Gomes 

Academic Editor

PLOS ONE